# MoGU: Mixture-of-Gaussians with Uncertainty-based Gating for Time Series Forecasting

## Abstract

We introduce Mixture-of-Gaussians with Uncertainty-based Gating (MoGU), a novel Mixture-of-Experts (MoE) framework designed for regression tasks and applied to time series forecasting. Unlike conventional MoEs that provide only point estimates, MoGU models each expert's output as a Gaussian distribution. This allows it to directly quantify both the forecast (the mean) and its inherent uncertainty (variance). MoGU's core innovation is its uncertainty-based gating mechanism, which replaces the traditional input-based gating network by using each expert's estimated variance to determine its contribution to the final prediction. Evaluated across diverse time series forecasting benchmarks, MoGU consistently outperforms single-expert models and traditional MoE setups. It also provides well-quantified, informative uncertainties that directly correlate with prediction errors, enhancing forecast reliability. Our code is available from: `https://anonymous.4open.science/r/moe_unc_tsf-65E1`.

## 1 Introduction

Mixture-of-Experts (MoE) is an architectural paradigm that adaptively combines predictions from multiple neural modules, known as "experts," via a learned gating mechanism. This concept has evolved from ensemble-based MoEs, where experts, jointly trained with a gating function, are often full, independent models whose outputs are combined to improve overall performance and robustness (Jacobs et al., 1991). More recently, MoE layers have been integrated within larger neural architectures, with experts operating in a latent domain. These "latent MoEs" offer significant scalability benefits, especially in large language models (LLMs) (Shazeer et al., 2017; Fedus et al., 2022). MoE makes it possible to train massive but efficient LLMs, where each token activates only a fraction of the model's parameters, enabling specialization, better performance, and lower computational cost compared to equally sized dense models.

Regardless of their specific implementation, conventional MoE systems typically produce point estimates, lacking a direct quantification of their uncertainty. In critical applications, this absence of uncertainty information hinders interpretability, making it difficult for users to gauge the reliability of a prediction and limits informed decision-making, as the system cannot express its confidence or identify ambiguous cases. Importantly, the learned gating mechanism, which dictates the relative contribution of each expert, does not take into account expert confidence, potentially leading to suboptimal routing decisions.

In this work, we propose Mixture-of-Gaussians with Uncertainty-based Gating (MoGU), a framework for uncertainty-aware MoE architectures, which provides explicit uncertainty quantification for both individual experts and the overall MoE model. Our approach fundamentally reimagines the expert's output: instead of a point estimate, we model each expert's prediction as a random variable drawn from a normal distribution. In this setup, each expert simultaneously predicts both the mean (the label estimate) and variance of the distribution, representing its predictive uncertainty. This shift enables a more nuanced understanding of expert behavior and the derivation of the overall model's uncertainty. Furthermore, we introduce a novel gating mechanism where the estimated uncertainty of each expert directly informs its relative contribution to the overall MoE prediction, bypassing the need for a

separate gating function typically found in traditional MoE setups. This creates a self-aware MoE where more confident experts naturally exert greater influence.

We evaluate MoGU on time series forecasting as our primary regression task. This choice is motivated by the inherent uncertainty in real-world time series data and the wide variety of expert architectures applicable to forecasting tasks across numerous domains (Lim & Zohren, 2021; Wang et al., 2024a). Our evaluation spans various expert types, forecasting benchmarks and forecasting horizon sizes, allowing for a comprehensive assessment of our method's efficacy. MoGU is shown to consistently yield more accurate forecasts compared to input-based gating MoE architectures, while simultaneously, providing uncertainty estimates that are positively correlated with prediction error. These estimates are available at both the individual expert and overall model levels. By further distinguishing between aleatoric (data-related) and epistemic (model-related) uncertainty, MoGU offers valuable insights into the source of a model's uncertainty. We also conducted a detailed ablation study to validate our key design choices.

In summary, our contributions are as follows:

- **MoGU: A Novel Framework for Uncertainty-Aware MoE Architectures**: We introduce a novel framework that directly quantifies uncertainty for both individual experts and the overall model, moving beyond conventional point estimates. A key innovation is a routing mechanism that uses each expert's estimated predictive uncertainty to dynamically determine its contribution to the final MoE output, replacing traditional input-based gating mechanisms.

- **MoGU Improves Time Series Forecasting**: Our method effectively reduces forecasting error across various benchmarks, horizon lengths, and expert architectures.

- **MoGU Provides Meaningful Uncertainty Estimates for Time Series Forecasting**: MoGU generates uncertainty estimates at the expert-level and overall. These estimates are positively correlated with prediction error, providing valuable insight into the model's confidence and the sources of its uncertainty.

By embedding uncertainty estimation into prediction and gating, MoGU moves beyond input-based gating MoEs toward architectures that are more accurate, transparent, and reliable.

## 2 RELATED WORK

**MoE Models** The pursuit of increasingly capable and adaptable artificial intelligence systems has led to the development of sophisticated architectural paradigms, among which the Mixture-of-Experts (MoE) stands out. MoE is an architectural concept that adaptively combines predictions from multiple specialized neural modules, often sharing a common architecture, through a learned gating mechanism. This paradigm allows for a dynamic allocation of computational resources, enabling models to specialize on different sub-problems or data modalities. Early implementations of MoE (Jacobs et al., 1991) focused on ensemble learning (ensemble MoE), where multiple models (experts) contributed to a final prediction. More recently, MoE layers have been seamlessly integrated within larger neural architectures, with experts operating in latent domains (latent MoE) (Shazeer et al., 2017; Fedus et al., 2022). This integration has proven particularly impactful in the realm of large language models (LLMs), where MoE layers have been instrumental in scaling models to unprecedented sizes while managing computational costs (Lepikhin et al., 2020; Jiang et al., 2024; Dai et al., 2024). By selectively activating only a subset of experts for each input token, MoEs enable models with vast numbers of parameters to achieve high performance without incurring the prohibitive inference costs of densely activated large models. Despite their contribution and adoption, both ensemble and latent MoE architectures typically output point estimates, both at the level of the individual expert and at the level of the overall model. This limits the ability to quantify uncertainty which is important for decision-making. Few works have explored uncertainty estimation for MoE architectures (see e.g. Pavlitska et al. (2025); Zhang et al. (2023)). In this work, we focus on ensemble MoE architectures, as uncertainty quantification is more directly applicable for decision making and interpretability. In our method, we view the experts of the MoE model as an ensemble of models that can be used to extract both aleatoric and epistemic uncertainties.

**Uncertainty Estimation for Regression Tasks.** Deep learning regression models are increasingly required not only to provide accurate point estimates but also to quantify predictive uncertainty. A large

body of research has focused on Bayesian neural networks, which place distributions over weights and approximate posterior inference using variational methods or Monte Carlo dropout, thereby producing predictive intervals (Gal & Ghahramani, 2016). Another line of work employs ensembles of neural networks to capture both aleatoric and epistemic uncertainties, with randomized initialization or bootstrapped training providing diverse predictions (Lakshminarayanan et al., 2017). More recently, post-hoc calibration techniques have been proposed, adapting classification-oriented approaches such as temperature scaling to regression settings, for instance by optimizing proper scoring rules or variance scaling factors (Kuleshov et al., 2018). Beyond probabilistic calibration, conformal prediction (CP) methods have gained attention due to their finite-sample coverage guarantees under minimal distributional assumptions. CP can be applied to regression to produce instance-dependent prediction intervals with guaranteed coverage, and has been extended to handle asymmetric intervals, distribution shift, and multi-target regression (Vovk et al., 2005; Romano et al., 2019).

**Time Series Forecasting and Uncertainty Estimation.** Time series forecasting is a critical discipline in machine learning and statistics, focusing on predicting future values from a sequence of historical data points ordered by time. This field has wide-ranging applications, including financial market analysis, energy consumption forecasting, weather prediction, and medical prognosis. Traditional statistical methods, such as Autoregressive Integrated Moving Average (ARIMA) and Exponential Smoothing, have been foundational. However, their effectiveness is often limited by their assumption of linearity and their inability to capture complex, non-linear dependencies. More recently, deep learning models, employing Transformers (Nie et al., 2023; Wu et al., 2021; Kitaev et al., 2020), Multi-Layer Perceptrons (MLPs) (Wang et al., 2024b; Zeng et al., 2023), and Convolutional Neural Networks (CNNs) (Wu et al., 2023), were shown to be effective in modeling temporal dynamics and long-range dependencies (Wang et al., 2024a; Lim & Zhoren, 2021; Wang et al., 2024c). The ability to quantify the uncertainty of a forecast, rather than providing just a single point estimate, is of paramount importance. Uncertainty quantification provides a confidence interval for the prediction, which is crucial for risk management and informed decision-making. Some recent works have introduced uncertainty estimation to time series forecasting (see e.g. Cini et al. (2025); Wu et al. (2025)). Given its wide-ranging applications, the importance of reporting uncertainty, and its challenging nature, time series forecasting serves as a highly suitable domain to evaluate the performance of MoGU.

## 3 METHOD

In this section, we introduce our uncertainty-based gating MoE framework. We begin by outlining the general formulation of MoE in Section 3.1. Subsequently, we present our proposed method, which extends this general MoE formulation to an uncertainty-based gating model, as detailed in Section 3.2. Finally, in Section 3.3, we demonstrate a concrete application of our mechanism to the task of time series forecasting.

### 3.1 INPUT-BASED GATING MIXTURE-OF-EXPERTS

A general formulation for an MoE network (Jacobs et al., 1991) can be defined as follows:

$$x \rightarrow (w_i(x), y_i(x)), \qquad i = 1, ..., k \tag{1}$$

where $x$ denotes the input, $y_i$ is the prediction of the $i$-th expert and $w_i$ is the weight the model assigns to that expert's prediction. The model's output is then calculated as the weighted sum of these expert predictions:

$$\hat{y} = \sum w_i(x) y_i(x). \tag{2}$$

Optimizing an MoE is achieved by minimizing the following loss:

$$\mathcal{L}_{\text{MoE}} = \sum w_i(x) \mathcal{L}(y_i(x), y) \tag{3}$$

where $y$ is the ground truth label and $\mathcal{L}$ is the loss function for the target task.

Typically, an MoE comprises a set of individual expert neural networks (often architecturally identical) that predict the outputs $y_i$, along with an additional gating neural module responsible for predicting the expert weights $w_i$. In its initial conception (Jacobs et al., 1991), both the experts and the gating module were realized as feedforward networks (the latter incorporating a softmax layer for

weight prediction). However, the underlying formulation is adaptable, and subsequent research has introduced diverse architectural implementations. Additionally, MoEs have also been implemented as layers within larger models (Shazeer et al., 2017), which we refer to as 'latent MoEs'.

### 3.2 MoGU: Mixture-of-Gaussians with Uncertainty-based Gating

We now describe our proposed framework, which extends MoEs to a Mixture-of-Gaussians with Uncertainty-based Gating (MoGU).

**From MoE to MoG.** We can add to each expert an uncertainty component that indicates how much the expert is confident in its decision:

$$x \rightarrow (w_i(x), y_i(x), \sigma_i^2(x)), \qquad i = 1, ..., k. \tag{4}$$

We can interpret $\sigma_i^2(x)$ as a variance term associated with the $i$-th expert. The experts' predictions and their variances can be jointly trained by replacing the individual expert loss $\mathcal{L}$ in Eq. (3) with the Gaussian Negative Log Likelihood (NLL) loss, denoted by $\mathcal{L}_{\text{NLLG}}$:

$$\mathcal{L}_{\text{MoG}} = \sum w_i(x) \mathcal{L}_{\text{NLLG}}(y; y_i(x), \sigma_i^2(x))) \tag{5}$$

with:

$$\mathcal{L}_{\text{NLLG}}(y; \mu, \sigma^2) = \frac{1}{2}(\log(\max(\sigma^2, \epsilon)) + \frac{(\mu - y)^2}{\max(\sigma^2, \epsilon)}) \tag{6}$$

where $\epsilon$ is used for stability. Similarly to the MoE formulation (Eq. (3)), the weights $w_i(x)$ are obtained through a softmax layer, which is computed by a separate gating module in addition to the experts given the input.

This model thus assumes that the conditional distribution of the labels $y$ given $x$ is an MoG. Therefore, at the inference step, the model prediction is given by:

$$\hat{y} = E(y|x) = \sum w_i(x) y_i(x) \tag{7}$$

and its variance is:

$$\text{Var}(y|x) = \underbrace{\sum w_i(x) \sigma_i^2(x)}_{\text{aleatoric uncertainty}} + \underbrace{\sum w_i(x)(\hat{y} - y_i(x))^2}_{\text{epistemic uncertainty}}. \tag{8}$$

The first term of (8) can be viewed as the *aleatoric uncertainty* and the second term is the *epistemic uncertainty* (see e.g. (Gal & Ghahramani, 2016)). Here, we use the experts and an ensemble of regression models (instead of extracting the ensemble from the dropout mechanism).

**From MoG to MoGU.** Once we add an uncertainty term for each expert, we can also interpret this term as the expert's relevance to the prediction task for the given input signal. We can thus transform the expert confidence information into relevance weights, allowing us to replace the standard input-based MoE gating mechanism, with a decision function that is based on expert uncertainties. We next present an alternative model, where the gating mechanism is based on using the variance of expert predictions as an uncertainty weight when combining the experts.

We can view each expert as an independently sampled noisy version of the true value $y$: $y_i \sim \mathcal{N}(y, \sigma_i^2(x))$. It can be easily verified that the maximum likelihood estimation of $y$ based on the experts' decisions $y_1, ..., y_k$ is:

$$\hat{y} = \arg\max_y \sum_i \log \mathcal{N}(y_i, ; y, \sigma_i^2) = \sum_i w_i y_i \tag{9}$$

s.t.

$$w_i = \frac{\sigma_i^{-2}}{\sum_j \sigma_j^{-2}}. \tag{10}$$

In other words, each expert is weighted in inverse proportion to its variance (i.e., proportional to its precision). In contrast to traditional MoEs where gating is learned as an auxiliary neural module, MoGU derives gating weights directly from uncertainty estimates, reframing expert selection as

probabilistic inference rather than an additional prediction task. We can thus substitute Eq. (10) in Eq. (5), to obtain the following loss function:

$$\mathcal{L}_{\text{MoGU}} = \sum_i \frac{\sigma_i^{-2}(x)}{\sum_j \sigma_j^{-2}(x)} \mathcal{L}_{\text{NLLG}}(y; y_i(x), \sigma_i^2(x))). \tag{11}$$

Further substituting (10) in (8) we obtain the variance reported by the MoGU model:

$$\text{Var}(y|x) = \underbrace{\frac{1}{\frac{1}{k}\sum_j \sigma_j^{-2}(x)}}_{\text{aleatoric uncertainty}} + \underbrace{\sum_i \frac{\sigma_i^{-2}(x)}{\sum_j \sigma_j^{-2}(x)}(\hat{y} - y_i(x))^2}_{\text{epistemic uncertainty}}. \tag{12}$$

Note that here the aleatoric uncertainty (the first additive term of (12)) is simply the harmonic mean of the variances of the individual expert predictions.

We provide a pseudo-code for MoGU in our Appendix as well as a complete PyTorch implementation to reproduce the results reported in our paper.

### 3.3 TIME SERIES FORECASTING WITH MoGU

We demonstrate the application of the MoGU approach to multivariate time series forecasting. The forecasting task is to predict future values of a system with multiple interacting variables. Given a sequence of $T$ observations for $V$ variables, represented by the matrix $x \in \mathbb{R}^{T \times V}$, the objective is to forecast the future values $y \in \mathbb{R}^{(T+h) \times V}$ where $h$ is the forecasting horizon.

Traditional neural forecasting models(forecasting 'experts') typically follow a two-step process. First, a neural module $g$, such as a Multi-Layer Perceptron (MLP) or a Transformer, encodes the input time series $x$ into a latent representation. Second, a fully connected layer $f$ regresses the future values $y$ from the latent representation $g(x)$. This process can be generally expressed as:

$$x \rightarrow f(g(x)). \tag{13}$$

To apply MoGU for time series forecasting, we need to extend forecasting experts with an uncertainty component as described in Eq. (4), by estimating the variance of the forecast in addition to the predicted values.

We implement this extension by introducing an additional MLP, denoted as $f'$, which predicts the variance $\sigma^2$ from the latent representation $g(x)$. The MLP $f'$ consists of a single hidden fully connected layer that maintains the same dimensionality as $g(x)$. The output of this layer is then passed through a Softplus function to ensure the variance is always non-negative and to promote numerical stability during training:

$$\sigma^2(x) = \log_2(1 + e^{f'(g(x))}). \tag{14}$$

The complete MoGU forecasting process is given by the following equation:

$$x \rightarrow (w_i, f_i(g_i(x)), \sigma_i^2(x)), \qquad i = 1, ..., k \tag{15}$$

where $w_i$ is computed as in Eq. (10) and $\sigma_i^2(x)$ is defined in Eq. (14).

## 4 EXPERIMENTS

We evaluate MoGU on several multivariate time series forecasting benchmarks. We compare its performance to the standard MoE, which lacks uncertainty estimation, and to a single-expert model. Our evaluation varies the number of experts, prediction horizon length, and expert architecture. The complete experimental setup is detailed in Section 4.1. The results of our evaluation are presented in Section 4.2.1. MoGU achieves competitive performance, consistently outperforming both the standard MoE and the single-expert models. We further analyze the reported uncertainty by our method in Section A.1. We find that the uncertainty estimates reported by MoGU are informative, positively correlated with prediction error, and accurately reflect the error trend. Finally, in Section

4.2.3, we present an ablation study that explores alternative design choices for our gating mechanism, loss, uncertainty head architecture, and the resolution at which uncertainty is reported. The results further validate the advantage of our proposed novel uncertainty-based gating and demonstrate the robustness of our framework.

## 4.1 Experimental setup

**Datasets.** We evaluate our method on eight widely used time series forecasting datasets (Wu et al., 2021): four Electricity Transformer Temperature (ETT) datasets (ETTh1, ETTh2, ETTm1, ETTm2) (Zhou et al., 2021), as well as Electricity[1], Weather[2], Exchange (Lai et al., 2018), and Illness (ILI)[3].

**Experimental Protocol.** Our experiments follow the standard protocol used in recent time series forecasting literature (Nie et al., 2023; Liu et al., 2023; Wang et al., 2024a). For the ILI dataset, we use a forecast horizon length $h \in \{24, 36, 48, 60\}$. For all other datasets, the forecast horizon length is selected from 96,192,336,720. A look-back window of 96 is used for all experiments. We report performance using the Mean Absolute Error (MAE) and Mean Squared Error (MSE). We evaluate the quality of our uncertainty estimates by computing the Pearson and Spearman correlation with respect to the prediction error. Specifically, for each individual variable, we correlate the model's reported uncertainty values with the corresponding MAE across all time points. We then average these correlation coefficients to get an overall measure.

**Expert Architecture.** MoGU is a general MoE framework compatible with various expert architectures. We evaluate it using three state-of-the-art expert models: iTransformer (Liu et al., 2023), PatchTST (Nie et al., 2023), and DLinear (Zeng et al., 2023). These models represent different architectural approaches, including Transformer and MLP-based designs.

**Implementation and Training Details.** We implemented MoGU in PyTorch (Paszke et al., 2019). For the expert architecture, we extended the existing implementations of PatchTST, iTransformer, and DLinear available from the Time Series Library (TSLib) (Wang et al., 2024a), to incorporate uncertainty estimation as detailed in Section 3.3. For training, we used a configuration similar to the one provided by TSLib. All models were trained for a maximum of 10 epochs with a patience of 3 epochs for early stopping. We used the Adam optimizer with a batch size of 8. The learning rate was set to $\lambda = 0.001$ for the Weather and Electricity datasets and $\lambda = 0.0001$ for all other datasets. All experiments were conducted on a single NVIDIA A100 80GB GPU.

Table 1: Multivariate forecasting results when using a single expert (standard forecasting setup) and when using MoE and MoG with Uncertainty-based gating (MoGU, ours), when varying on the number of experts. The best MAE and MSE results for for a 96-time point horizon are shown in bold for each dataset.

| Configuration | Single Expert | MoE | | | | MoGU (ours) | | | |
|---|---|---|---|---|---|---|---|---|---|
| Num. Experts | 1 | 2 | 3 | 4 | 5 | 2 | 3 | 4 | 5 |
| ETTh1 | 0.398 | 0.391 | 0.393 | 0.398 | 0.392 | 0.385 | 0.380 | 0.382 | **0.381** |
| ETTh2 | 0.295 | 0.307 | 0.299 | 0.305 | 0.311 | 0.284 | **0.283** | 0.286 | 0.286 |
| ETTm1 | 0.341 | 0.349 | 0.332 | 0.347 | 0.339 | 0.320 | 0.320 | 0.314 | **0.312** |
| ETTm2 | 0.188 | 0.186 | 0.179 | 0.180 | 0.177 | 0.179 | 0.179 | 0.176 | **0.175** |

## 4.2 Results

### 4.2.1 Time Series Forecasting with MoGU

Table 1 compares MoGU's performance against single-expert and standard MoE configurations on the ETT datasets. Using iTransformer as the expert architecture and varying the number of experts from 2 to 5, MoGU consistently yields more accurate predictions than both single-expert and standard MoE settings. Tables 2 and 3 provide further comparisons between a three-expert MoE and MoGU.

---

[1]https://archive.ics.uci.edu/ml/datasets/ElectricityLoadDiagrams20112014

[2]https://www.bgc-jena.mpg.de/wetter/

[3]https://gis.cdc.gov/grasp/fluview/fluportaldashboard.html

Table 2: Multivariate long-term forecasting results with MoE and MoGU (ours), with iTransformer and PatchTST as the expert architectures. We report the MAE and MSE for each configuration, using prediction lengths $L \in \{24, 36, 48, 60\}$ for the ILI dataset and $L \in \{96, 192, 336, 720\}$ for the others. The best MSE results for each configuration are shown in bold.

| Expert | | iTransformer | | | | PatchTST | | | |
|---|---|---|---|---|---|---|---|---|---|
| Mixture Type | | MoE | | MoGU (ours) | | MoE | | MoGU (ours) | |
| Metric | | MAE | MSE | MAE | MSE | MAE | MSE | MAE | MSE |
| ETTh1 | 96 | 0.410 | 0.393 | **0.400** | **0.380** | **0.406** | **0.386** | 0.415 | 0.409 |
| | 192 | 0.432 | 0.437 | **0.431** | **0.436** | 0.448 | 0.459 | **0.443** | **0.453** |
| | 336 | 0.472 | 0.504 | **0.454** | **0.479** | 0.465 | 0.485 | **0.459** | **0.484** |
| | 720 | **0.489** | **0.500** | 0.491 | 0.501 | 0.494 | 0.510 | **0.483** | **0.485** |
| ETTh2 | 96 | 0.348 | 0.299 | **0.336** | **0.283** | 0.347 | 0.298 | **0.331** | **0.277** |
| | 192 | 0.396 | 0.377 | **0.387** | **0.361** | 0.400 | 0.375 | **0.386** | **0.357** |
| | 336 | 0.427 | **0.413** | 0.425 | 0.415 | 0.440 | 0.422 | **0.423** | **0.406** |
| | 720 | 0.447 | 0.435 | **0.442** | **0.421** | 0.460 | 0.443 | **0.447** | **0.426** |
| ETTm1 | 96 | 0.367 | 0.332 | **0.356** | **0.320** | 0.371 | 0.337 | **0.362** | **0.326** |
| | 192 | 0.396 | 0.382 | **0.379** | **0.363** | 0.398 | **0.380** | **0.393** | 0.389 |
| | 336 | 0.411 | 0.407 | **0.404** | **0.400** | 0.407 | **0.400** | 0.407 | 0.400 |
| | 720 | 0.460 | 0.500 | **0.438** | **0.466** | 0.448 | 0.465 | **0.442** | **0.460** |
| ETTm2 | 96 | 0.261 | **0.179** | **0.260** | **0.179** | 0.264 | 0.177 | **0.259** | **0.175** |
| | 192 | 0.306 | 0.246 | **0.302** | **0.245** | 0.308 | 0.247 | **0.303** | **0.242** |
| | 336 | 0.345 | 0.307 | **0.339** | **0.301** | **0.346** | **0.304** | 0.346 | 0.307 |
| | 720 | 0.401 | 0.403 | **0.395** | **0.397** | 0.405 | 0.408 | **0.403** | **0.405** |
| ILI | 24 | 0.864 | 1.786 | **0.827** | **1.756** | 0.866 | 1.871 | **0.822** | **1.848** |
| | 36 | 0.882 | 1.746 | **0.825** | **1.629** | 0.875 | 1.875 | **0.835** | **1.801** |
| | 48 | 0.948 | 1.912 | **0.843** | **1.634** | 0.878 | **1.798** | **0.844** | 1.818 |
| | 60 | 0.979 | 1.986 | **0.881** | **1.692** | 0.904 | 1.864 | **0.864** | **1.831** |
| Weather | 96 | 0.253 | 0.208 | **0.249** | **0.207** | 0.237 | 0.196 | **0.230** | **0.188** |
| | 192 | **0.283** | **0.246** | **0.283** | 0.251 | 0.268 | 0.235 | **0.265** | **0.232** |
| | 336 | **0.315** | **0.296** | 0.317 | 0.300 | 0.308 | 0.291 | **0.303** | **0.287** |
| | 720 | **0.361** | **0.369** | **0.361** | 0.371 | 0.353 | 0.363 | **0.351** | **0.361** |
| Electricity | 96 | **0.235** | **0.144** | 0.238 | 0.148 | **0.248** | **0.161** | 0.257 | 0.169 |
| | 192 | 0.254 | **0.162** | **0.251** | 0.163 | **0.258** | **0.170** | 0.263 | 0.179 |
| | 336 | **0.269** | **0.175** | 0.269 | 0.179 | **0.276** | **0.188** | 0.286 | 0.200 |
| | 720 | **0.297** | **0.204** | 0.302 | 0.216 | **0.314** | **0.231** | 0.319 | 0.242 |
| Num. Wins | | 4 | 9 | **21** | **18** | 5 | 8 | **21** | **19** |

Table 3: Multivariate forecasting results for MoE and MoGU (ours), using DLinear, iTransformer, and PatchTST as expert architectures. The best MAE and MSE results for each configuration are shown in bold for a 96-time point horizon.

| Expert | DLinear | | | | iTransformer | | | | PatchTST | | | |
|---|---|---|---|---|---|---|---|---|---|---|---|---|
| Mixture Type | MoE | | MoGU (ours) | | MoE | | MoGU (ours) | | MoE | | MoGU (ours) | |
| Metric | MAE | MSE | MAE | MSE | MAE | MSE | MAE | MSE | MAE | MSE | MAE | MSE |
| Exchange | 0.213 | 0.086 | **0.209** | **0.080** | 0.225 | 0.010 | **0.208** | **0.089** | **0.201** | 0.086 | 0.202 | **0.084** |
| ETTh1 | **0.400** | **0.382** | **0.400** | **0.382** | 0.410 | 0.393 | **0.400** | **0.380** | 0.406 | 0.386 | 0.415 | 0.409 |
| ETTh2 | 0.373 | 0.320 | **0.366** | **0.308** | 0.348 | 0.299 | **0.336** | **0.283** | 0.347 | 0.298 | **0.331** | **0.277** |
| ETTm1 | **0.360** | **0.322** | 0.363 | 0.338 | 0.367 | 0.332 | **0.356** | **0.320** | 0.371 | 0.337 | **0.362** | **0.326** |
| ETTm2 | 0.285 | 0.189 | **0.271** | **0.183** | 0.261 | **0.179** | **0.260** | **0.179** | 0.264 | 0.177 | **0.259** | **0.175** |

MoGU outperforms standard MoE in the majority of cases across different multivariate forecasting datasets and horizon lengths, utilizing iTransformer, PatchTST, and DLinear as expert architectures.

### 4.2.2 UNCERTAINTY QUANTIFICATION FOR TIME SERIES FORECASTING WITH MOGU

To assess how well MoGU's reported uncertainty aligns with its actual prediction errors, we compute the Pearson (R) and Spearman ($\rho$) correlation coefficients between them. Table 4 presents these coefficients for the aleatoric, epistemic, and total uncertainties (as defined in Eq. 12).

Table 4: Pearson (R) correlation and Spearman ($\rho$) coefficients between the uncertainty reported by MoGU and the MAE of its predictions. The correlations were computed per variable and then averaged, showing the relationship for a 96-time point horizon. All reported results are statistically significant with a p-value $\leq 0.00001$.

| Uncertainty | | Aleatoric (A) | | Epistemic (E) | | Total (A+E) | |
|---|---|---|---|---|---|---|---|
| Corr. Coeff. | | R | $\rho$ | R | $\rho$ | R | $\rho$ |
| iTransfor. | ETTh1 | 0.25 | 0.22 | 0.03 | 0.04 | 0.25 | 0.22 |
| | ETTh2 | 0.15 | 0.20 | 0.08 | 0.15 | 0.15 | 0.21 |
| | ETTm1 | 0.27 | 0.29 | 0.10 | 0.13 | 0.27 | 0.30 |
| | ETTm2 | 0.15 | 0.17 | 0.13 | 0.24 | 0.16 | 0.19 |
| PatchTST | ETTh1 | 0.26 | 0.23 | 0.05 | 0.05 | 0.26 | 0.23 |
| | ETTh2 | 0.14 | 0.17 | 0.12 | 0.20 | 0.14 | 0.17 |
| | ETTm1 | 0.31 | 0.30 | 0.07 | 0.11 | 0.31 | 0.30 |
| | ETTm2 | 0.11 | 0.11 | 0.14 | 0.25 | 0.11 | 0.11 |

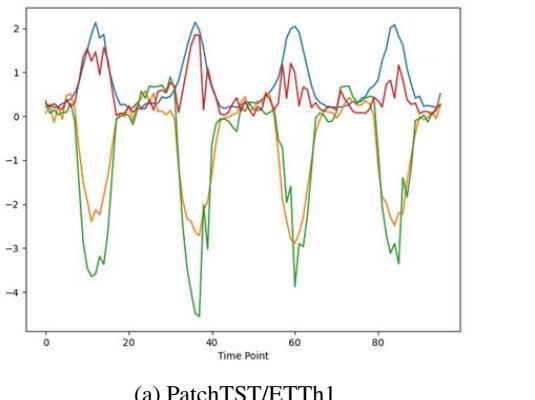 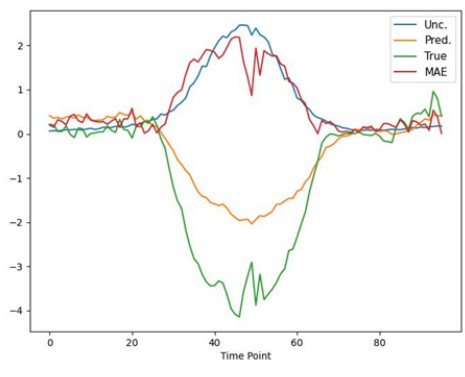

(a) PatchTST/ETTh1        (b) iTransformer/ETTm1

Figure 1: Example forecasts along with the ground truth, the MAE and uncertainty reported by MoGU with three experts. The forecasts for the Etth1 dataset (a) were generated using PatchTST as the expert architecture, while those for Ettm1 (b) were generated using iTransformer.

Table 5: Forecasting errors for a 96-time point horizon of MoE, MoG and MoGU models. The best results are shown in bold for each configuration and dataset.

| Mixture Type | | MoE | | MoG | | MoGU | |
|---|---|---|---|---|---|---|---|
| Metric | | MAE | MSE | MAE | MSE | MAE | MSE |
| iTransfor. | ETTh1 | 0.410 | 0.393 | 0.403 | 0.387 | **0.400** | **0.380** |
| | ETTh2 | 0.348 | 0.299 | 0.340 | 0.288 | **0.336** | **0.283** |
| | ETTm1 | 0.367 | 0.332 | 0.360 | 0.326 | **0.356** | **0.320** |
| | ETTm2 | 0.261 | 0.179 | **0.256** | **0.175** | 0.260 | 0.179 |
| PatchTST | ETTh1 | **0.406** | **0.386** | 0.420 | 0.413 | 0.415 | 0.409 |
| | ETTh2 | 0.347 | 0.298 | 0.343 | 0.291 | **0.331** | **0.277** |
| | ETTm1 | 0.371 | 0.337 | 0.372 | 0.337 | **0.362** | **0.326** |
| | ETTm2 | 0.264 | 0.177 | **0.259** | 0.176 | **0.259** | **0.175** |

We observe a statistically significant positive correlation between MoGU's uncertainty estimates and the Mean Absolute Error (MAE) of its predictions. Interestingly, the correlation with aleatoric uncertainty is typically higher than with epistemic uncertainty. Since aleatoric uncertainty represents the inherent randomness in the data itself, this correlation suggests that the model can use uncertainty estimates to identify data points where irreducible randomness makes accurate predictions difficult, thereby leading to higher errors.

Fig. 1 illustrates the relationship between MoGU's prediction error and uncertainty estimates by showing the predicted and ground truth values alongside the MAE and reported uncertainty for

Table 6: Ablation study of the uncertainty head's architecture with a 96-time point horizon. The best results are shown in bold for each configuration and dataset.

| Head Architecture | | FC | | MLP | |
|---|---|---|---|---|---|
| | Metric | MAE | MSE | MAE | MSE |
| iTransfor. | ETTh1 | **0.399** | 0.383 | 0.400 | **0.380** |
| | ETTh2 | 0.338 | 0.286 | **0.336** | **0.283** |
| | ETTm1 | 0.357 | 0.321 | **0.356** | **0.320** |
| | ETTm2 | 0.261 | 0.178 | 0.260 | 0.179 |
| PatchTST | ETTh1 | 0.410 | 0.401 | **0.415** | **0.409** |
| | ETTh2 | 0.340 | 0.285 | **0.331** | **0.277** |
| | ETTm1 | 0.356 | **0.320** | 0.362 | 0.326 |
| | ETTm2 | 0.260 | **0.174** | **0.259** | 0.175 |

representative examples. The uncertainty at each time point closely follows the prediction error. We further show the Pearson correlation heatmaps in Fig. 2 in our Appendix. These heatmaps further visualize the relationship between the Mean Absolute Error (MAE) of MoGU's predictions and its reported uncertainties (aleatoric, epistemic, and total), when using MoGU with three iTransformer experts. The analysis is presented per variable for each of the ETT datasets, highlighting the extent to which different uncertainty components correlate with predictive error. While the correlation between uncertainty and MAE varies among variables, it remains consistently positive.

### 4.2.3 ABLATIONS

We conducted an ablation study to evaluate our key design choices. For all experiments, we used a configuration with three experts.

**Gating Mechanism.** Table 5 compares our MoGU to a standard input-based gating mechanism (Jacobs et al., 1991), when employed by a deterministic MoE and with a MoG. The input-based method utilizes a separate neural module to predict weights by processing the input before a softmax layer. We evaluated the MoE, MoG and MoGU methods on four ETT datasets using iTransformer and PatchTST as the expert architectures. Our uncertainty-based gating consistently resulted in a lower prediction error.

**Uncertainty Head Architecture.** We also evaluated the design of our uncertainty head, which is implemented as a shallow Multi-Layer Perceptron (MLP) with a single hidden fully connected layer. Table 6 compares this to an alternative using only a single fully connected layer. The MLP alternative performed better in most cases, though the performance difference was relatively small.

**Resolution of Uncertainty Estimation.** Table 7 in our Appendix explores an alternative where the expert estimates uncertainty at the variable level ('Time-Fixed'), rather than for each individual time point ('Time-Varying'). Predicting uncertainty at the higher resolution of a single time point yielded better results, demonstrating the advantage of our framework's ability to provide high-resolution uncertainty predictions. We note that our framework is flexible and supports both configurations.

Additional ablations for our **Loss Function** are provided in the Appendix (Section A.2).

## 5 CONCLUSION

We introduced MoGU, a novel extension of MoE for time series forecasting. Instead of using traditional input-based gating, MoGU's gating mechanism aggregates expert predictions based on their individual uncertainty (variance) estimates. This approach led to superior performance over single-expert and conventional MoE models across various benchmarks, architectures, and time horizons. Our results suggest a promising new direction for MoEs: integrating probabilistic information directly into the gating process for more robust and reliable models.

**Limitations and Future Work:** While MoGU shows promise for time series forecasting, broadening its scope to other regression (and classification) tasks, will further validate its robustness and generalization. In addition, adapting its dense gating for sparse architectures like those in LLMs remains a challenge for future work.

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

## A APPENDIX

We provide additional results and details that were not included in the main text due to space limitations.

### A.1 CORRELATION HEATMAPS: UNCERTAINTY VERSUS PREDICTION ERROR

Fig. 2 shows correlation heatmaps discussed in Section in the main text. This heatmap visualizes the relationship between the Mean Absolute Error (MAE) of MoGU's predictions and its reported uncertainties (aleatoric, epistemic, and total) for a model using three iTransformer experts.

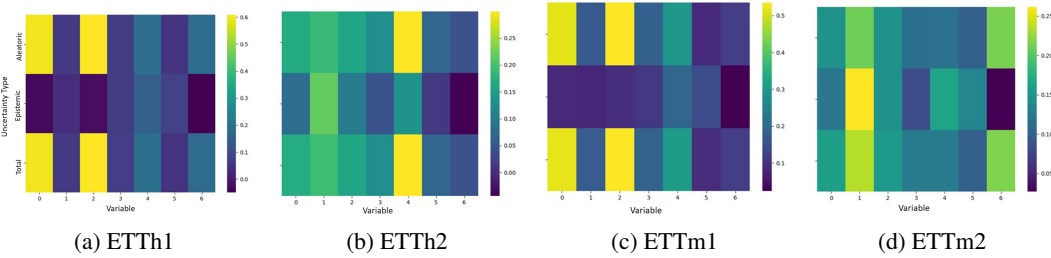

|        (a) ETTh1        |        (b) ETTh2        |        (c) ETTm1        |        (d) ETTm2        |

Figure 2: Heatmaps of the Pearson correlation between MoGU's reported uncertainties (aleatoric, epistemic, and total) and the MAE of its predictions. The correlation is displayed per variable for the ETT datasets.

## A.2 ADDITIONAL ABLATIONS

**Resolution of Uncertainty Estimation.** We provide Table 7, discussed in the main text. This table explores an alternative where the expert estimates uncertainty at the variable level ('Time-Fixed'), rather than for each individual time point ('Time-Varying'). **Loss Function.** We note that the MoGU model can also be optimized through the following MoG loss:

$$\mathcal{L} = -\log(\sum_i w_i(x)\mathcal{N}(y; y_i(x), \sigma_i^2(x))) \tag{16}$$

where $\mathcal{N}$ is the Normal density function and the loss has the form of a Negative Log Likelihood (NLL) of a MoG distribution. We compare the performance of our model when using the loss presented in Eq. 5 and when using the aforementioned alternative (Eq. 16). The results of this experiment, presented in Table 8 in our Appendix, suggest that optimizing with our proposed loss (Eq. 5) yields more effective learning and consistently better results by imposing a stricter constraint on expert learning compared to the MoG loss.

Table 7: Ablation study on the resolution of reported uncertainty. We compare two methods for estimating variance in both MoE and MoGU: estimating it once per horizon versus estimating it for each time point (per variable in both cases). The table reports the MAE and MSE for each configuration. All results were generated using iTransformer as the expert architecture with a 96-time-point horizon.

| Prediction Variance | Time-Fixed | | Time-Varying | |
|---|---|---|---|---|
| Metric | MAE | MSE | MAE | MSE |
| ETTh1 | 0.401 | 0.392 | **0.400** | **0.380** |
| ETTh2 | 0.337 | 0.290 | **0.336** | **0.283** |
| ETTm1 | 0.360 | 0.324 | **0.356** | **0.320** |
| ETTm2 | **0.255** | **0.174** | 0.260 | 0.179 |

Table 8: Ablation study of MoGU's loss. We compare the loss formulation in Eq. 5, used by MoGU to an alternative MoG loss, given in Eq. 16

| Loss Formulation | Eq. 16 (Alt. MoG loss) | | Eq. 5 (MoGU's loss) | |
|---|---|---|---|---|
| Metric | MAE | MSE | MAE | MSE |
| 96 | 0.343 | 0.304 | **0.336** | **0.283** |
| 192 | 0.389 | 0.378 | **0.387** | **0.361** |
| 336 | **0.424** | 0.422 | 0.425 | **0.415** |
| 720 | **0.438** | **0.421** | 0.442 | 0.421 |

## A.3 MOGU'S ALGORITHM

We provide the pseudo code for MoGU in Listing 1 to enhance clarity. Furthermore, to ensure reproducibility, our code and the scripts needed to reproduce the main results are available at: `https:`

**Algorithm 1** Mixture-of-Gaussians with Uncertainty-based gating (MoGU)

---

**Require:** Training set $X$, labels $y$
**Ensure:** Model parameters $\theta$
1: **for** each training epoch **do**
2:      **for** each mini-batch $\mathcal{B}$ **do**
3:          **for** each sample $x \in \mathcal{B}$ **do**
4:              **for** each expert $i = 1, \ldots, k$ **do**
5:                  Compute expert output $f_i(x) = \mathcal{N}(y; \mu_i(x, \theta), \sigma_i^2(x, \theta))$.
6:                  Set $w_i(x) = \frac{\sigma_i^{-2}(x)}{\sum_j \sigma_j^{-2}(x)}$.
7:              **end for**
8:          **end for**
9:          Compute loss $\mathcal{L} = \sum w_i(x)\mathcal{L}_{\mathcal{NLLG}}(y; y_i(x), \sigma_i^2(x)))$.
10:          Update model parameters.
11:      **end for**
12: **end for**
13: Test time prediction is $\hat{y}(x) = \sum_i w_i(x)\mu_i(x)$.
14: Test time prediction uncertainty is: $\sum_i w_i(x)\sigma_i^2(x) + \sum_i w_i(x)(\hat{y}(x) - \mu_i(x))^2$.

---

`//anonymous.4open.science/r/moe_unc_tsf-65E1` We implemented MoGU to be highly configurable, so that users can specify the number of experts, the expert architecture, the mixture type (MoE or MoG) and the gating mechanism.

