# OpenReview forum: "MoGU: Mixture-of-Gaussians with Uncertainty-based Gating for Time Series Forecasting"
_ICLR.cc/2026/Conference — ICLR 2026 Conference Withdrawn Submission_

### Official Review · Reviewer_cXJp · 2025-10-26

**Soundness:** 2
**Presentation:** 3
**Contribution:** 2
**Rating:** 4
**Confidence:** 4

**Summary:**

This paper proposes a framework called mixture-of-Gaussian with uncertainty based gating. The framework combines a mixture of Gaussian framework with a gating mechanism, which which uses the variance of each experts to determine their contribution to the final prediction. Experimental results show that the proposed approach has competitive performance against the generic mixture of experts approach.

**Strengths:**

- The method is straightforward and the paper is easy to follow
- The simplicity and effectiveness of the method as an add-on module suggest that it could potentially be adapted into existing methods to improve their performance
- The numerical results show that the proposed framework is competitive against the generic MoE method
- An ablation study is performed to support the choices of model components

**Weaknesses:**

- A main concern is the limited novelty: The proposed approach appears to be a simple extension from the long-established MoE method, where each expert is assumed to be Gaussian, and their weights are determined by the variance.
- For an ICLR paper, given this limited novelty, I would expect a more detailed theoretical analysis, but the theoretical contribution is quite limited
- In the numerical experiments, although state-of-the-art expert architectures are used as feature extractors, the method is only compared against the generic MoE model. Thus, it is not clear how well the proposed approach performs against more modern MoE frameworks
- Continuing from above, do the MoE and proposed approach outperform the standalone expert architectures (as originally proposed, without modification for incorporating MoE)?
- It is claimed as a key contribution that the proposed approach provides meaningful uncertainty estimates, as the mean and variance of the prediction distribution can be obtained. However, only point forecasting metrics are evaluated in the experiments. And probabilistic forecasting metrics are not considered. Thus, the probabilistic aspect of the method is not evaluated

**Questions:**

See weaknesses

---

### Official Review · Reviewer_vWhc · 2025-10-26

**Soundness:** 3
**Presentation:** 3
**Contribution:** 2
**Rating:** 4
**Confidence:** 4

**Summary:**

This paper introduce MoGU, an extension of MoE for time series forecasting.  MoGU’s gating mechanism aggregates expert predictions based on their individual uncertainty (variance) estimates and surpasses MOE on multiple datasets.

**Strengths:**

1. The authors propose MOGU, a Mixture-of-Experts (MoE) architecture equipped with uncertainty-aware gating. On top of the standard MoE framework, MOGU introduces a routing mechanism that leverages estimated uncertainty for expert selection. Compared with conventional MoE structures, the model can ensemble inputs across multiple experts according to each expert’s uncertainty level.
2. MOGU demonstrates superior performance over standard MoE models on time series tasks. Models adopting the MOGU architecture consistently outperform their MoE counterparts across multiple datasets and various expert backbones.

**Weaknesses:**

In Contribution 3, the authors claim that MOGU can effectively estimate uncertainty in time-series forecasting tasks. However, I find this assertion unconvincing, as the proposed uncertainty modeling only concerns each expert’s individual predictive variance rather than the intrinsic uncertainty within the temporal sequence itself. Consequently, the estimation of prediction error cannot be directly derived from the experts’ own uncertainty estimates.

**Questions:**

Please refer to the weakness. In addition,
1. The authors should clarify how the prediction error can be derived from the MoE experts’ uncertainty estimates, and provide additional metrics and analyses to substantiate this claim.
2. If error estimation is indeed feasible, then for multivariate time series, the model should account for inter-variable dependencies, modeling predictive uncertainty with a covariance structure rather than independent variances.
3. On the Electricity dataset, MOGU performs notably worse than MoE, suggesting that MOGU may not generalize well to certain data types; the authors should provide a clear explanation for this performance gap.

---

### Official Review · Reviewer_Wfhr · 2025-10-29

**Soundness:** 3
**Presentation:** 3
**Contribution:** 2
**Rating:** 4
**Confidence:** 4

**Summary:**

The paper proposes MoGU, a MoE variant where each expert outputs a Gaussian. At inference and in training, experts are precision-weighted rather than using a learned input-gating network; this yields a closed-form gating rule and a natural decomposition of predictive uncertainty into aleatoric and epistemic components. The method is instantiated for multivariate time-series forecasting and evaluated on eight benchmarks, showing modest but consistent accuracy gains over single-expert and standard input-gated MoE, plus positive correlations between reported uncertainty and realized error. Ablations cover gating choice, variance-head architecture, time-varying vs. time-fixed variance, and an alternative MoG loss.

**Strengths:**

- Simple, principled gating—drop-in replacement for input-gated MoE; easy to implement.
- Broad applicability across expert architectures.
- Empirical coverage: eight datasets, three expert families, multiple horizons, and several ablations.
- Provides uncertainty breakdown (aleatoric vs. epistemic) with statistically significant positive correlation to errors.

**Weaknesses:**

- Probabilistic metrics absent: no NLL/CRPS, correlation with error alone doesn’t establish calibration or decision utility.
- Baselines: lacks comparisons to (i) deep ensembles with heteroscedastic heads, (ii) mixture density networks with learned mixture weights, (iii) conformal/quantile forecasting pipelines; current baseline is mostly MoE with learned gate.
- Objective-gating coupling: experts both predict $\sigma^2$ and are weighted by $\sigma^{-2}$, which can incentivize variance shrinkage; beyond $\epsilon$-clipping in the NLL, there’s no explicit regularizationto prevent pathological gating.
- Independence assumption across experts: precision weighting is optimal if expert errors are independent/unbiased; correlated experts violate this, suggesting a GLS-style weighting ($\Sigma^{-1}$) would be more appropriate—unaddressed.
- Compute/training budget appears small (≤10 epochs), possibly handicapping stronger baselines.

**Questions:**

- Please report at least CRPS, or empirical coverage of, say, 50/80/90% Gaussian intervals. Do MoGU’s intervals calibrate better than MoE?
- Variance collapse: How do you guard against $\sigma^2 \rightarrow 0$ dominating the gate?
- Expert correlation: Have you measured inter-expert error correlation? If substantial, can you approximate GLS weights (e.g., using a low-rank covariance across experts) or at least add diversity-encouraging perturbations?
- Granularity of gating: With time-varying $\sigma^2$, are gates computed per target dimension/time step or aggregated to a scalar $w_i$?
- Broader baselines: Add deep ensembles (heteroscedastic), MDN with learned mixture weights, and conformalized quantile models; also include NLL/CRPS for those.
- Fair training budgets: Can you increase training epochs/batches or report wall-clock matched training to ensure baselines are not under-tuned?

---

### Official Review · Reviewer_pmA8 · 2025-10-31

**Soundness:** 3
**Presentation:** 2
**Contribution:** 2
**Rating:** 4
**Confidence:** 4

**Summary:**

This work presents MoGU, a Gaussian-output extension of the Mixture of Experts (MoE) framework. The core idea is to have each expert output a Gaussian distribution and then aggregate them using a precision-weighted gating network, where the weight for each expert is proportional to the inverse of its variance (\(w_i \propto \sigma_i^{-2}\)). This setup naturally yields a closed-form decomposition of the total predictive uncertainty into aleatoric and epistemic components. The authors have conducted extensive evaluations on 8 time-series datasets with 3 different backbone architectures, comparing against several strong baselines.

**Strengths:**

* The theoretical derivation is clean and easy to follow, especially the progression from standard MoE to the Gaussian-output variant.
* The precision-weighting gating scheme is well-motivated from a maximum likelihood perspective (Eqs. 9-10), which is a solid theoretical foundation.
* The closed-form uncertainty decomposition (Eq. 12) is a nice property, making the interpretation of model uncertainty straightforward.
* The experimental setup is comprehensive, covering multiple datasets and model architectures, which makes the empirical validation quite convincing.
* The results in Table 3 show consistent improvements over standard MoE, and the ablation studies in Table 5 effectively demonstrate the advantage of uncertainty-based gating.

**Weaknesses:**

* While technically sound, the core methodological innovation feels somewhat incremental, building directly on established MLE principles for precision weighting.
* The calibration analysis is rather limited - only reporting correlation with MAE doesn't fully validate the uncertainty quantification. Proper calibration metrics like interval coverage or CRPS are needed.
* The statistical rigor could be improved: no multiple random seeds are reported, and the training protocol (10 epochs max with early stopping) seems somewhat aggressive and might lead to unstable comparisons.
* Resource reporting and experimental fairness: Details like tuning budgets, training time, and computational costs are missing, making it hard to assess the practical efficiency.
* There's a potential data issue in Table 3 - the MSE of 0.010 for iTransformer on Exchange seems unusually low compared to other values.
* Reproducibility is a concern as the code repository appears incomplete (only README visible).
* The writing could benefit from better flow between sections, particularly in the method description.
* The "uncertainty head" is mentioned but not properly defined in the methods section, leaving implementation details unclear.

**Questions:**

1. Beyond MAE correlation, could you provide proper calibration metrics (empirical coverage, reliability diagrams, or CRPS) to validate that experts with smaller variance indeed deserve higher weights?
2. Table 5 suggests uncertainty gating helps, but to really pin down the benefit: compared to a strong input-gating baseline with expert variance prediction, what's the net gain? Could you run this comparison with multiple seeds and significance testing?
3. I'm curious about the robustness of your uncertainty decomposition - does Eq. 12 rely on strong conditional independence or Gaussian assumptions? What happens if some experts are poorly calibrated? Could this bias the gating mechanism?
4. The training seems quite brief (10 epochs max). Have you checked if this might disadvantage baselines that converge slower? Some learning curves would help address this concern.
5. In Table 3, the 0.010 MSE for iTransformer on Exchange looks suspicious - could this be a formatting error? Also, could you clarify how "Num. Wins" is computed across different experimental conditions?
6. Regarding Eqs. 5-6: I'd appreciate some clarification on the originality. Are these your novel contributions? If building on prior work, please cite appropriately. Also, could you discuss why you chose this formulation over the exact MoG NLL?
7. The uncertainty head seems crucial but is under-specified. Could you formally define its architecture, how it ensures non-negative outputs, where it plugs into the backbone, and how it's trained?

---

### Note · Authors · 2026-01-12

I have read and agree with the venue's withdrawal policy on behalf of myself and my co-authors.